# Effect of a Concurrent Cognitive Task, with Stabilizing Visual Information and Withdrawal, on Body Sway Adaptation of Parkinsonian’s Patients in an Off-Medication State: A Controlled Study

**DOI:** 10.3390/s20185059

**Published:** 2020-09-06

**Authors:** Arnaud Delafontaine, Clint Hansen, Iris Marolleau, Stefan Kratzenstein, Arnaud Gouelle

**Affiliations:** 1CIAMS, University Paris-Sud, Université Paris-Saclay, 91405 Orsay, France; iris.marolleau@gmail.com; 2CIAMS, Université d’Orléans, 45067 Orléans, France; 3Department of Neurology, University Hospital Schleswig-Holstein, Christian-Albrechts-Universität zu Kiel, 24098 Kiel, Germany; c.hansen@neurologie.uni-kiel.de; 4CAU Motion Lab, Kiel University, Olshausenstraße 74, 24098 Kiel, Germany; Stefan.Kratzenstein@email.uni-kiel.de; 5Institute of Sport Science, Kiel University, Olshausenstraße 74, 24098 Kiel, Germany; 6ProtoKinetics, Havertown, PA 19083, USA; arnaud.gouelle@gmail.com; 7Laboratory Performance, Santé, Métrologie, Société (PSMS), UFR STAPS, 51100 Reims, France

**Keywords:** Parkinson’s, body sway, center of pressure, cognitive task, visual deprivation, dual-task

## Abstract

*Background*: In persons with Parkinson’s disease (pwPD) any additional somatosensory or distractor interference can influence the posture. When deprivation of vision and dual-task are associated, the effect on biomechanical performance is less consistent. The aim of this study was to evaluate the role of the visual deprivation and a cognitive task on the static balance in earlier stage PD subjects. *Methods:* Fifteen off-medication state pwPD (9 women and 6 men), 67.7 ± 7.3 years old, diagnosed PD since 5.4 ± 3.4 years, only Hoehn and Yahr state 2 and fifteen young control adults (7 women and 8 men) aged 24.9 ± 4.9 years, performed semi-tandem task under four randomized experimental conditions: eyes opened single-task, eyes closed single-task, eyes opened dual-task and eyes closed dual-task. The center of pressure (COP) was measured using a force plate and electromyography signals (EMG) of the ankle/hip muscles were recorded. Traditional parameters, including COP pathway length, ellipse area, mediolateral/anteroposterior root-mean-square and non-linear measurements were computed. The effect of vision privation, cognitive task, and vision X cognitive was investigated by a 2 (eyes opened/eyes closed) × 2 (postural task alone/with cognitive task) repeated-measures ANOVA after application of a Bonferroni pairwise correction for multiple comparisons. Significant interactions were further analyzed using post-hoc tests. *Results*: In pwPD, both COP pathway length (*p* < 0.01), ellipse area (*p* < 0.01) and mediolateral/anteroposterior root-mean-square (*p* < 0.01) were increased with the eyes closed, while the dual-task had no significant effect when compared to the single-task condition. Comparable results were observed in the control group for who COP pathway was longer in all conditions compared to eyes opened single-task (*p* < 0.01) and longer in conditions with eyes closed compared to eyes opened dual-task (*p* < 0.01). Similarly, all differences in EMG activity of pwPD were exclusively observed between eyes opened vs. eyes closed conditions, and especially for the forward leg’s soleus (*p* < 0.01) and backward tibialis anterior (*p* < 0.01). *Conclusions*: These results in pwPD without noticeable impairment of static balance encourage the assessment of both visual occlusion and dual-task conditions when the appearance of significant alteration during the dual-task could reveal the subtle worsening onset of the balance control.

## 1. Introduction

Parkinson’s disease (PD) is a neurodegenerative disease leading to plastic changes in the primary motor cortex, with a multifaced physiopathology, whose cardinal motor symptoms are bradykinesia, rigidity, rest tremor and postural instability [1]. Postural balance requires constant shifts in the center of gravity and neuromuscular responses to maintain within the limits of stability. Most patients complain of abnormal postural deficits including impairment of sensory integration, postural reflexes, reduction of limits of stability, instability and difficulty reacting to external challenges or unexpected perturbations [2]. However, at an earlier stage, assessment using a low difficulty balance task could underestimate perturbations or responsiveness to medication.

The semi-tandem stance, a position where the toes of one foot are in line with the inside arch of the other foot, has been found to be the most challenging quiet stance task that can be sustained by healthy young and older subjects for at least 30 s [3]. Significantly fewer subjects (i.e., 56% [3] were capable to conduct more challenging upright quiet stances, for example, tandem stance). Postural control is more asymmetric in individuals in early or moderate stages [4] of PD compared to neurologically healthy individuals, because PD typically starts on one side of the body and remains asymmetric throughout the disease in most people [5]. Therefore, a semi-tandem stance could be challenging enough to reveal abnormalities and compensatory mechanisms in the less symptomatic persons living with PD (pwPD). Closing the eyes also increases the difficulty of the task and leads to poorer postural balance compared to holding the position with the eyes open [6]. Marchese et al. (2003) [7] showed a postural deficit in both control subjects and pwPD. Performing cognitive or attention-demanding tasks (e.g., counting backwards by series of 3s or reciting the days of the week backwards) during quiet standing can further demonstrate effects on body sway [8]. The interaction between cognitive processes and postural control has been repeatedly shown in healthy adults [9] and pwPD [7,8,10]. Postural balance performance is poorer in pwPD when associated with an additional task [10] and the effects are significantly more evident in patients with a history of prior falls [7]. However, when Holmes et al. [11] tested the impact of various levels of task complexity, the most complex cognitive task resulted in significantly less postural sway in pwPD than in healthy controls, suggesting an over control of postural adjustments during the most complex task. To investigate the effects of a cognitive dual-task on balance performance in pwPD, the medication state must also be considered. Levodopa’s effect on static balance remains controversial [12,13]. Both increase and decrease of the sway area during quiet standing have been reported after levodopa intake during quiet standing [14]. While understanding the differences between the ON and OFF-medication states is very important for follow-up in PD, the investigation of differences between pwPD with low balance impairment during their OFF-medication state and controls could highlight balance-specific PD markers, which potentially help to identify pwPD who are still not diagnosed. Electromyographical (EMG) activities are modified during reactive balance responses in pwPD [15]. Recording of leg muscles during an off-medication state (i.e., after 12 h withdrawal from anti-Parkinson’s medications, as defined by Defer et al. [16]) can potentially identify specific neuromuscular mechanisms involved in balance impairment and selective markers for exercise-based rehabilitation, too. Indeed, while Levodopa has been shown to reduce the activity of the M. tibialis anterior during walking [17], limited data is available for balance tasks.

This study was set to test how visual deprivation and dual-task affect body sway in OFF-state pwPD who hold a semi-tandem stance. A control group of healthy subjects underwent the same protocol. We expected an effect of both visual deprivation and dual-task on the semi-tandem performances, especially in the patients. We hypothesized a significant increase in neuromuscular activity of the leg muscles during the more challenging condition, in relation with an increase of ankle contribution to maintain body sways [18].

## 2. Materials and Methods

### 2.1. Subjects

Fifteen pwPD and fifteen healthy young adults (Table 1) participated in this cross-sectional study. The inclusion criteria for the pwPD were original diagnosis of PD by a neurologist based on standard clinical criteria [19]; ability to walk without walking aids (i.e., without a walker or a cane); Hoehn and Yahr [20] state 2 only (bilateral involvement without impairment of balance); no history of falls; no anti-parkinsonian medications intake for 12 h prior to the study session (i.e., “OFF-medication state”); Mini-Mental State Examination [21] score above 26, which considered as the cut-off for normal cognitive function. Restriction to pwPD with Hoehn and Yahr at state 2 was set to involve persons who did not demonstrate impairment of the static balance. Exclusion criteria were defined as visual, hearing or orthopaedic impairments; other neurological disorders, dementia, severe dyskinesia, medical diagnosis of dementia, history or presence of psychotic episode since the PD diagnosis, deep brain stimulator implant.

All subjects provided informed written consent as required by the Declaration of Helsinki. The experiment was approved by the local ethics committee of the University Paris-Saclay (EA4532) and by the “Comité de Protection des Personnes Ile-de-France XI” under identification number 19028-60429.

### 2.2. Experiments

Clinical assessments of pwPD were conducted at least 30 min before the study procedure. These tests included the Unified Parkinson’s Disease Rating Scale motor examination [22], Berg Balance Scale, 15 s sit-to-stand movements and were conducted by an experienced neuro-physiotherapist. The Berg Balance Scale was used to confirm the absence of balance impairments during a single-leg-stance test which is, in pwPD, clinically correlated to an important stage of disease progression with significant worsening of postural stability [23].

All subjects (i.e., both pwPD and healthy young adults) had to stand for 60 s on a force platform with their feet in a semi-tandem position (Figure 1) with the toes of the non-dominant foot in line with the inside arch of the dominant foot. The foot dominance [24] has been determined with the mobilizing or manipulating limb as the dominant foot, and the supporting limb as the non-dominant limb.

The subjects completed three trials for each of the four randomly-assigned conditions: semi-tandem only with eyes open (EO-ST) and eyes closed (EC-ST), semi-tandem with additional cognitive task with eyes open (EO-DT) and eyes closed (EC-DT). In the eyes open conditions (EO-ST and EO-DT), they were instructed to fix the gaze on a 10 cm diameter target placed at eye level at 6 m distance.

The subjects sat for 3 min between the trials to prevent them from extensive fatigue. The feet outline was drawn on the platform to ensure identical foot positions between all measurements. During the mentally idle condition, instructions such as “concentrate on the task” [25,26], or “make deliberate efforts to reduce body sway” were avoided. The cognitive task consisted in subtracting three to a series of numbers given by the experimenter. During the whole duration of the trial, this one continuously enounced random number between 1 and 100. The participant had to repeat the number then to give the resulting value after subtracting three. The subjects were instructed to perform the task as best as they could and to continue counting regardless of any errors they may have made.

### 2.3. Instrumentation and Signal Analyses

The ground reaction forces and moments were acquired using a floor-embedded force plate, sampling at 1000 Hz. Due to the inter-centeral study design the measurements were conducted with different systems but identical data collection settings: AMTI BP9001800 (0.9 × 1.8 m, AMTI, Watertown, MA, USA) for the pwPD and a Kistler 9260AA (Kistler, Winterthur, Switzerland) for the young adults.

The positions of the center of pressure (COP) in the mediolateral (ML) and anteroposterior (AP) directions were computed offline using a custom-made MATLAB (MathWorks, Natick, MA, USA) routine and were bandpass filtered between 0.1 and 5 Hz [27] with a no-lag 2nd order Butterworth filter. The COP trajectories were computed for each of the three trials and then averaged.

To quantify the differences between the conditions, the following variables were computed from postural sways: COP pathway length (PL), ellipse area (EA), root-mean-square (RMS), mean frequency (MeanFreq) and median frequency (MedianFreq). The RMS was defined as the quadratic mean and is a measure of the sway displacement in both the AP and ML directions [mm]. The PL was defined as the total length of the COP trajectory on the platform [mm]. MeanFreq came from the power spectrum of the COP sway in both AP and ML directions. The EA represents the area of the smallest ellipse which covered 95% of the COP samples [mm^2^] [3].

Three non-linear methods—Sample Entropy (SampEn), Multi-Scale Entropy (MSE), and Multivariate Multi-Scale Entropy (MMSE)-were used to measure the regularity of the COP signals from the unfiltered signal and to compare their sensitivity between the incremented time series and the original time series. SampEN is essentially a negative logarithm of the conditional probability of the sequences of a data vector. If a vector of length N has repeated itself in tolerance γ for m points, it will also do so for m + 1 point. The conditional probability means the ratio of counts of repeated time of m + 1 point to that of m points. Thereby, high SampEN arises from a low probability of repeated sequences in the data. Higher SampEN means lower regularity and more complexity in the data. Based on SampEN, MSE is a method evaluating the complexity of signals over different time scales while MMSE generalizes the analysis to the multivariate case [26]. In short, the three non-linear entropy methods are effective in measuring the complexity of time series, specifically the SampEN for the univariate vector, MSE for the univariate vector in a multiple time-scale, and MMSE for the multivariate matrix in multiple time-scales, respectively. The complexity index, defined as the integral of the MSE or MMSE curve, was used for the statistical analysis of the entropy measures (MSE, MMSE).

EMG activities were analysed for the M. tibialis anterior (TA), M. soleus (SOL), and M. Tensor of Fascia Lata (TFL). According to the respective leg positions, Muscle were determined as frontal and backward TA, SOL and TFL. Signals were recorded using a wireless EMG system in the pwPD (Zero-Wire, Aurion Ltd., Milan, Italy) and in the young adults (Myon AG, Schwarzenberg, Switzerland). The standard guidelines recommended by the SENIAM [28] were applied to prepare the skin and to identify the correct positions the sensors. Bipolar pre-gelled Ag-AgCl electrodes with 1.5 cm space between recording leads were used to record EMG activity at a sampling frequency of 1000 Hz. After bandpass filtering between 20 and 450 Hz on the EMG signals and rectification, the RMS were computed for each single trial, and then averaged. To measure the level of performance for the cognitive task, the number of correct answers given during the trial and the correct response percentage were logged.

### 2.4. Statistical Comparisons

Descriptive statistics were computed for all variables under the four conditions. The normality of distribution (Shapiro-Wilk test) and homoscedasticity of data (Levene’s test) were verified before the application of the parametric tests. Since the criteria were not satisfied for all parameters, sway metrics were log-transformed to meet the normality request for further analysis. In pwPD and controls, the effect of vision privation, cognitive task, and vision X cognitive were judged by a 2 (eyes opened/eyes closed) × 2 (without/with cognitive task) repeated-measures ANOVA. Significant interactions were further analyzed using Bonferroni post-hoc tests. Also, a Bonferroni pairwise correction was applied to account for multiple comparisons (i.e., EO-ST, EC-ST, EO-DT, EC-DT) and a threshold of *p* < 0.01 was selected to consider significant results.

## 3. Results

The characteristics of the pwPD and participants in the control group are displayed in Table 1. None of the participants had impairments in one of the first nine items of the Berg Balance Scale (Table 1), the ninth being assessment of the single-leg balance ability. A Score of 0 for the first item of UPDRS III, which is related to speech, provided evidence that the participants had no issue with articulation that would contribute to modification in COP measures [29].

In pwPD (Table 2 and Table 3), differences have been identified between both eyes open (EO-ST and EO-DT) and eyes closed conditions (EC-ST and EC-DT) for COP pathway (*p* < 0.001, longer with eyes closed: EO-ST 1606 ± 1060 mm vs. EC-ST 2546 ± 930 mm, EO-DT 2105 ± 1749 mm vs. EC-DT 2874 ± 1776 mm), ellipse area (*p* < 0.001, larger with EC: EO-ST 101 ± 56 mm^2^ vs. EC-ST 283 ± 168 mm^2^, EO-DT 159 ± 118 mm^2^ vs. EC-DT 296 ± 177 mm^2^), RMS of AP and ML amplitude (*p* < 0.001, larger with EC). No differences were found between other conditions for the other parameters, including non-linear metrics. Regarding muscle activities, forward SOL and backward TA resulted in increased RMS with eyes closed than with eyes open (*p* < 0.01), while the RMS of backward SOL and forward TFL were higher during EC-DT than during EO-ST (*p* < 0.01). In addition, multiple regressions have been conducted with age, sex, UPDRS and Berg Balance Scale as potential confounders. No effects were observed.

The analyses of the control group lead to differences in the COP pathway length and ellipse area. The COP pathway was longer in all conditions compared to EO-ST (*p* < 0.01, EO-ST 1654 ± 220 mm, EC-ST 1985 ± 223 mm, EO-DT 1793 ± 272 mm, EC-DT 2041 ± 332 mm) and longer with eyes closed compared to EO-DT (*p* < 0.0001). Ellipse area was larger with eyes closed compared to EO-ST (*p* ≤ 0.01). Differences were found in MSE for AP and ML amplitudes (*p* ≤ 0.01), demonstrating a lower regularity for the eyes closed conditions than during EO-ST. EMG activities did not differ across the four conditions.

No differences between pwPD and controls were found for the length of the COP pathway, MeanFreq AP and few conditions for MeanFreq ML and MedianFreq ML. The other parameters were significantly different between groups, with better outcomes for the controls (Table 3).

## 4. Discussion

In this study we investigated challenging postural tasks in pwPD with no noticeable balance impairments, to determine whether, and to which extend, subtle balance disorders can be detected. The current protocol, where visual deprivation and dual-task paradigms were evaluated during semi-tandem stance, provided new insights. Overall trends in the results were similar in pwPD and young controls, although COP path length was significantly longer for controls when performing dual-task than single-task with the eyes opened. Maintaining postural stability involves sensorimotor transformations that continuously integrate several sensory inputs and coordinate motor outputs to muscles throughout the body [30]. In this context, vision plays a major role in postural control and contributes to regulating postural stability during quiet standing [31]. Our results did not contradict this statement nor previous results examining the tandem stance in pwPD [32], where visual deprivation increased postural instability, as evidence by higher COP displacement, velocity, ellipse area and RMS, in comparison to open eyes condition.

The lack of visual inputs implies a greater reliance on the integration of the other sensory outputs, with a suitable informational processing speed. A healthy person relies on visual, somatosensory, and vestibular information for balance control, but all or some of the components of this system may be dysfunctional in Parkinsonian patients [33]. Therefore, any impairment in postural reflexes and timing responses can further affect standing performance in pwPD. Visual deprivation may also reinforce internalization of the focus.

Vuillerme and Nafati [34] observed a worsening in postural stability when control adults were asked to intentionally focus on their body sways. Their results suggested that the internal focus promoted the use of less automatic control process and hampered the efficiency for controlling posture during quiet standing. Again, issues in conscious motor control, which can underly Parkinson’s disease even at early stages for balance impairments, might alter global performance during quiet semi-tandem standing.

On the other hand, all difference in EMG activity have been exclusively observed between eyes opened vs. eyes closed conditions, especially for the forward leg’s M. soleus and backward M. tibialis anterior, which demonstrated increased RMS when eyes were closed. This result was expected based on the M. soleus’ fundamental support role during quiet standing [31], with a well-established relationship between anteroposterior sway of the COP and the integrated EMG of the M. soleus [31]. It also suggests that asymmetric weight distribution in a semi-tandem position might be exacerbated when the eyes are closed. With two separate force-plates, one under each foot, Barbieri et al. [35] found greater postural asymmetry in pwPD compared to controls in a semi-tandem position, considering this to reflect either the unilateral development of motor symptoms caused by the asymmetric degeneration of dopaminergic neurons in the substantia nigra [5], or to result from a compensatory strategy. Indeed, the increase in postural asymmetry during more complex task in pwPD [36] could promote a strategy where one leg manages weight support while the other controls sway.

More surprising were the results under the dual-task condition. The pwPD included in the study demonstrated no alteration to biomechanical outcomes or EMG RMS when a cognitive task was added to the postural primary task, independently of the vision condition. The pwPD have issues in automating movements, which increases the attentional demand during daily activities and generates difficulties associating a simultaneous cognitive task to a motor task [37]. Nieuwhof et al. [38] found poorer performance in pwPD performing dual-task than in those performing a single-task. The authors hypothesized that this might result from a loss of functional segregation between neighboring striatal territories which occurs specifically in a dual-task context. In our results, different explanations could have contributed to no significant changes under dual-task. First, the cognitive task provides an external focus which diverts attention from body sways and reduces conscious control [39]. Some studies have shown the potential of external focus to reduce postural instability during quiet standing compared to an internal focus in pwPD [40] and to improve postural instability when pwPD took dopamine medication [41]. However, if we did not find significant difference due to the dual-task, confidence interval higher limit was higher for COP pathway and ellipse area under dual-task. When some pwPD could have benefited from an internal focus effect, it appears also that others had more difficulties during the dual-task. As we asked the participants to perform the cognitive task as best as they can, this might have generated a task prioritization leading to more automatic control [42] or stiffness of ankle for postural control. In the current study, pwPD were assessed during an OFF-medication state and we can wonder whether we would have seen improvement in dual-task during an ON state. Based on recent results by Workman and Thrasher [43], it would not be the case. During a similar protocol to ours, sixteen pwPD completed single- and dual-task standing with eyes open and eyes closed for 3 min each in off and on medication states, to investigate if dopaminergic medication improved standing balance automaticity during a phoneme monitoring dual-task. No change suggesting an improvement in automaticity was found during dual-task and a negative effect was even seen under medication. Dual-task paradigms are undertaken to challenge attentional capacities. Two main theorical models propose to rationalize the mechanisms which induce interferences in one or both tasks. The capacity sharing model considers attention as a limited resource and implies a sharing of attention between the two tasks. If both tasks requirement overpasses the attentional capacity, then the achievement of the primary task must be done to the detriment of the secondary task by drawing on the attentional load that it brings into play [44]. On the other hand, the bottleneck theory [45] suggests that interference occurs when two tasks compete for the access to an attentional mechanism whose capacity is limited to the management of a single central operation at a time. To complete one task, processing of the second task is temporarily postponed, resulting in performance decrements in the second task. Nonappearance of difference between single-task and dual-task could be explained by an insufficiently challenging cognitive task for attentional resources of our participants (capacity sharing) or because the second task was not in direct competition with the standing task (bottleneck).

In the most complex task, under dual-task with eyes closed, the increase in both M. soleus and M. tibialis anterior activity of the backward leg might have reflected the necessity for the pwPD to use an ankle strategy with co-activation of the muscles and could underly the postural instability during quiet standing [46], since this phenomenon is independently associated with dynamic postural control abilities [47]. However, stiffness induced by this co-contraction could have deleterious effects and repercussions in time either by transfer to other movements, such as the gait initiation process (i.e., where the M.soleus of the trailing limb also plays an important role in maintaining dynamic postural stability by actively braking the vertical fall of the center of mass, see [48]), and/or through depletion of selective control capacities at this level. In a rehabilitation perspective, integration of dual-tasks in a semi-tandem position as an exercise in pwPD in an early stage could be worthwhile and promote higher M. tibialis anterior activity of the backward leg, especially during an OFF-medication state, when Levodopa administration has been shown to reduce M. tibialis anterior EMG amplitude during gait [17].

In addition to the most standard COP metrics, we analyzed non-linear parameters, such as entropy, to leverage information that traditional measurements do not reveal. While the use of non-linear methods has a long tradition to distinguish between visual conditions in elderly patients [49], former results on COP regularity revealing the amount of attention invested in posture [50] are in line with our findings. Entropy measures are usually univariate time series analyses and combining anterior-posterior and medio-lateral sway directions to extend the entropy measures to a multivariate case may still be helpful when dealing with postural sway. Among the studied entropy measures, only MMSE showed significant differences between the eyes open and eyes closed condition. Higher entropy values are related to more irregularity, which may be associated with a functional decline of the postural control system and consequently maladaptive responses to perturbations leading to altered balance control [51]. Previously, Hansen et al. [26] showed that manipulating the original time series and computing entropy measures can distinguish between subjects with high postural capacity, although the question arises as to how much signal processing is actually needed if even classical postural parameters can distinguish between kinesthetic conditions in pwPD. Particularly with the specific parameters and properties of the non-linear measures as entropy, values are based on a coarse-graining procedure with specific tolerance settings (i.e., constant fraction of the variance of the original time series). Entropy measures assess the complexity of physiological time series signals rather than measure motor performance which is complex to understand from a clinical perspective.

Some limitations must be discussed. The findings around dual-task are hard to interpret. First, the cognitive performance during the dual-task was not assessed and the trade-off between cognitive and balance tasks cannot be determined. Then, the findings may be due to the mild disease of the participants, or perhaps cortical control is not as important in early disease or compared to other motor tasks such as gait. One recent review suggests the importance of cerebellum for postural control [52]. Regarding the group of controls, we did not compare pwPD to age-matched adults. Due to current restriction (i.e., COVID-19), the access to healthy participants was highly limited. This was overcome by measuring young adults instead, in a different laboratory, and it also explained the different force plate and EMG equipment. However, the data analysis was conducted by the same person using the same algorithm. For this reason, the results from the control group were used here as a reference for an optimal functioning, to see if the trends were the same under blinded condition and/or dual-task, but not for a direct comparison with the pwPD. Finally, as mentioned previously, the use of a single, large force-plate did not allow to investigate further differences between the forward and backward legs.

Future studies should first overcome these limitations. The use of a conceptual model would be helpful to characterize patterns of cognitive-motor dual-task interference [53], especially when, as suggested by Workman and Thrasher [43], some symptoms in PD might simulate stability in some posturographic measures and affect interpretation.

## 5. Conclusions

In pwPD without noticeable impairment of static balance, visual deprivation seems to alter more standing performance during semi-tandem stance than the addition of a secondary cognitive task. However, for clinical assessment, it can be valuable to combine both visual occlusion and dual-task conditions, when studies in pwPD with higher functional impairments usually demonstrated significant poorer balance on dual-task than single-task [8]. Appearance of significant alteration during dual-task could reveal subtle worsening onset of the balance control.

## Figures and Tables

**Figure 1 sensors-20-05059-f001:**
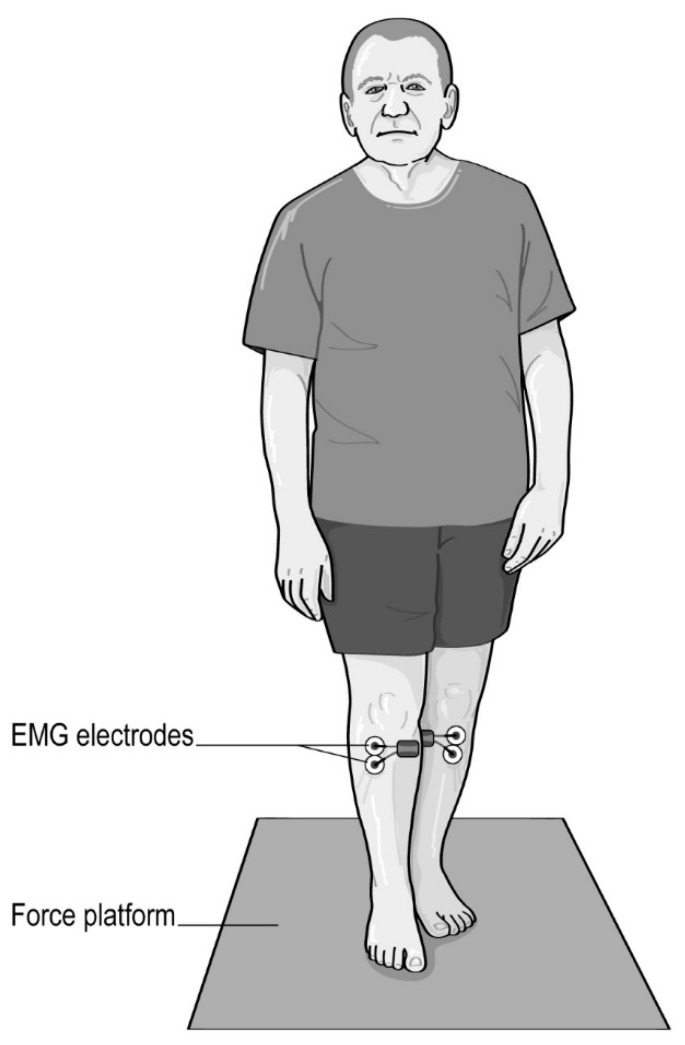
Semi-tandem stance with eyes opened.

**Table 1 sensors-20-05059-t001:** Demographic characteristics of patients with Parkinson’s disease and the control group.

	Patients with PD Mean (SD) [CI 95%]	Control Group Mean (SD) [CI 95%]	Between-Group Comparison *p*-Value
**Individuals (n)**	15	15	
**Age (years)**	67.7 (7.3) [63.6–71.7]	24.9 (4.9) [22.1–27.6]	***p* < 0.01**
**Height (m)**	1.67 (0.08) [1.62–1.71]	1.77 (8.4) [1.72–1.82]	***p* < 0.01**
**Weight (kg)**	67.8 (15.8) [59.1–76.5]	69.8 (12.2) [63.0–76.5]	***p >* 0.05**
**Body Mass Index (kg/m^2^)**	24.0 (3.8) [22.0–26.1]	22.2 (2.4) [20.8–23.5]	***p >* 0.05**
**Gender (F/M)**	9/6	7/8	***p >* 0.05**
**Dominant leg (L/R)**	3/12	2/13	***p >* 0.05**
**UPDRS motor examination**	24.5 (10.1) [18.9–30.0]		-
**UPDRS III Speech (item 3.1)**	0 (0)		-
**Sit-to-Stand in 15 s (n)**	8.1 (1.8) [7.1–9.1]		-
**Mini-Mental State Examination**	29.1 (1.5) [28.2–29.9]		-
**Berg Balance Scale (BBS)**	55.1 (2.1) [53. 9–56.2]		-
**Points in first 9 items of BBS (points)** **Disease Durations (years)**	36 (0) [36–36] 5.4 (3.4) [3.5–7.3]		- -
**Hoehn & Yahr Scale**	2 (0)		-

F female; L left; M male; R right.

**Table 2 sensors-20-05059-t002:** Summary statistics (Mean [CI 95%]) for COP parameters, electromyographic RMS and cognitive performance in pwPD.

	Eyes Open Single-Task	Eyes Open Dual-Task	Eyes Closed Single-Task	Eyes Closed Dual-Task
**COP Pathway (mm)**	1606 [933–2280]	2105 [930–3279]	2546 [1954–3139] *§	2874 [1745–4002] *§
**Ellipse Area 95% (mm^2^)**	101 [67–134]	159 [84–234]	283 [181–384] *§	296 [190–403] *§
**RMS ML (mm)**	3.7 [2.9–4.4]	4.3 [3.2–5.5]	6.5 [5.2–7.7] *§	6.2 [5.2–7.3] *§
**RMS AP (mm)**	1.4 [1.1–1.7]	1.9 [1.2–2.5]	2.5 [2.0–2.9] *§	2.5 [1.9–3.2] *§
**MeanFreq ML (Hz)**	0.011 [0.009–0.012]	0.011 [0.009–0.012]	0.010 [0.010–0.011]	0.011 [0.009–0.012]
**MedianFreq ML (Hz)**	0.018 [0.015–0.021]	0.018 [0.015–0.021]	0.017 [0.016–0.018]	0.018 [0.015–0.021]
**Mean Freq AP (Hz)**	0.010 [0.009–0.012]	0.011 [0.009–0.012]	0.010 [0.010–0.011]	0.011 [0.009–0.012]
**MedianFreq AP (Hz)**	0.018 [0.015–0.021]	0.018 [0.015–0.021]	0.017 [0.016–0.018]	0.018 [0.015–0.021]
**SampEn ML**	1.25 [1.23–1.27]	1.24 [1.21–1.27]	1.25 [1.23–1.27]	1.25 [1.23–1.27]
**SampEn AP**	1.26 [1.23–1.28]	1.25 [1.22–1.28]	1.26 [1.24–1.28]	1.26 [1.24–1.29]
**MSE ML**	6.05 [5.24–6.87]	6.39 [5.31–7.46] *	6.09 [5.35–6.83] *	6.33 [5.42–7.24] *
**MSE AP**	6.91 [6.03–7.79]	7.28 [6.27–8.28]	6.96 [6.06–7.86] *	7.10 [6.12–8.07] *
**MMSE**	20.52 (18.47–22.57]	20.26 [18.12–22.41]	20.95 [18.81–23.10]	21.17 [19.17–23.17]
**RMS Soleus-Forward (mV)**	955 [356–1284]	998 [370–1267]	1082 [396–1301] *	1111 [432–1352] *
**RMS Soleus-Backward (mV)**	1248 [551–1625]	1319 [732–1667]	1333 [632–2233]	1517 [779–2195] *
**RMS Tibialis Ant.-Forward (mV)**	1324 [367–2801]	1543 [408–2817]	1738 [615–3371]	1679 [478–3662]
**RMS Tibialis Ant.-Backward (mV)**	816 [312–2291]	844 [249–1945]	1334 [512–2416] *§	1204 [348–1967] *§
**RMS TFL-Forward (mV)**	613 [346–1048]	657 [330–1179]	685 [359–1119]	777 [345–1330] *
**RMS TFL-Backward (mV)**	746 [224–2103]	668 [265–1367]	842 [235–2343]	807 [265–2423]
**Correct answers (n)**		23.2 [18.9–27.5]		21.7 [18.3–25.2]
**Percentage of correct answers (%)**		88.9 [83.3–94.6]		89.1 [84.1–94.0]

**Abbreviations:** COP: Center of Pressure; RMS: Root Mean Square; ML: Mediolateral; AP: Anteroposterior: MeanFreq: Mean Frequency; MedianFreq: Median Frequency; SampEn: Sample Entropy; MSE: Multi-Scale Entropy; MMSE: Multivariate Multi-Scale Entropy; TFL: Tensor of Fascia Lata. **Symbols:** * significantly different from EO-ST; § significantly different from EO-DT (*p* < 0.0125).

**Table 3 sensors-20-05059-t003:** Significant differences between the four conditions in the control group using Bonferroni post hoc tests. Results are considered as significant when *p* < 0.0125.

**COP Pathway**	EO-ST	EO-DT	EC-ST	EC-DT
EO-ST		**0.0021**	**<0.0001**	**<0.0001**
EO-DT	**0.0021**		**0.0001**	**<0.0001**
EC-ST	**<0.0001**	**0.0001**		1.0000
EC-DT	**<0.0001**	**<0.0001**	1.0000	
**Ellipse Area**	EO-ST	EO-DT	EC-ST	EC-DT
EO-ST		1.0000	**0.0044**	**0.0040**
EO-DT	1.0000		**0.0117**	**0.0107**
EC-ST	**0.0044**	**0.0117**		1.0000
EC-DT	**0.0040**	**0.0107**	1.0000	
**MSE ML**	EO-ST	EO-DT	EC-ST	EC-DT
EO-ST		**0.0105**	**0.0103**	**0.0004**
EO-DT	**0.0105**		1.0000	0.5911
EC-ST	**0.0103**	1.0000		0.6011
EC-DT	**0.0004**	0.5911	0.6011	
**MSE AP**	EO-ST	EO-DT	EC-ST	EC-DT
EO-ST		0.0552	**0.0056**	**0.0012**
EO-DT	0.0552		1.0000	0.4000
EC-ST	**0.0056**	1.0000		1.0000
EC-DT	**0.0012**	0.4000	1.0000	

**Abbreviations:** EO: Eyes Opened; EC: Eyes Closed; ST: Single-task; DT: Dual-task; COP: Center of Pressure; RMS: Root-mean-square; ML: Mediolateral; AP: Anteroposterior; MSE: Multi-Scale Entropy.

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
