# Peer review of "Effect of a Concurrent Cognitive Task, with Stabilizing Visual Information and Withdrawal, on Body Sway Adaptation of Parkinsonian’s Patients in an Off-Medication State: A Controlled Study"

_sensors, 2020, doi:10.3390/s20185059_

Round 1

Reviewer 1 Report

The manuscript by Delafontaine et al., aimed to assess the impact of dual task and vision on postural sway in people with mild Parkinson’s disease compared to controls. This is an interesting article which will benefit the field, a few minor revisions have been suggested prior to publication.

  • Line 50 ‘postural balance performance is less effective for pwPD when balance is associated with an additional task’. Perhaps this should read ‘postural balance performance is poorer in pwPD when associated with an additional task’
  • Could the authors display the data in Table 3 in a more concise format? Perhaps only significant data could be displayed with the rest of the data in supplementary material?
  • The inclusion/exclusion criteria stated that those with an MMSE below 26 were excluded from the study. This may have impacted on the dual task performance, were many participants excluded using this criteria?
  • Could the authors provide further rationale for why they tested off medication and describe how this may have impacted on findings?
  • The findings around dual task are hard to interpret. The findings may be due to the mild disease of the participants, or perhaps cortical control isn’t as important in early disease or compared to other motor tasks such as gait. One recent review suggests the importance of cerebellum for postural control https://doi.org/10.1016/j.neubiorev.2020.04.028. It has been noted in the limitations that cognitive performance during the dual task was not assessed and therefore the trade-off between cognitive and balance tasks cannot be determined.
  • Statistically there are a lot of multiple comparisons alongside a small number of participants, the authors should be more stringent and control for multiple comparisons.

Author Response

Reviewer 1 

The manuscript by Delafontaine et al., aimed to assess the impact of dual task and vision on postural sway in people with mild Parkinson’s disease compared to controls. This is an interesting article which will benefit the field, a few minor revisions have been suggested prior to publication.

R1 - Comment#1

Line 50 ‘postural balance performance is less effective for pwPD when balance is associated with an additional task’. Perhaps this should read ‘postural balance performance is poorer in pwPD when associated with an additional task’.

Authors:

As requested, we have changed the term.

Page 3, Lines 54-55: “Postural balance performance is poorer in pwPD when associated with an additional task [10]”.

R1 - Comment#2

Could the authors display the data in Table 3 in a more concise format? Perhaps only significant data could be displayed with the rest of the data in supplementary material?

Authors: Non-significant results have been removed from Table 3.

R1 - Comment#3

The inclusion/exclusion criteria stated that those with an MMSE below 26 were excluded from the study. This may have impacted on the dual task performance, were many participants excluded using this criteria?

Authors: It was maybe not clear in the first draft, but the MMSE>26 was an inclusion criterium. We intentionally selected persons without cognitive impairments avoid it to be an additional possible effect. Therefore, nobody has been excluded due to a lower MMSE. Inclusion/Exclusion section has been written in an improved manner.

Page 3, Lines 73-79: “The inclusion criteria for the pwPD were as follows: original diagnosis of PD by a neurologist based on standard clinical criteria [19]; to be able to walk without walking aids (i.e. without a walker or a cane); Hoehn and Yahr [20] state 2 only (bilateral involvement without impairment of balance); no fall history; no antiparkinsonian medications taken for 12 hours prior to the study session (i.e. “OFF-medication state”); Mini-Mental State Examination [21] score above 26, which considered as the cut-off for normal cognitive function. Restriction to pwPD with Hoehn and Yahr at state 2 was set to involve persons who did not demonstrate impairment of the static balance. Patients were not included if they presented any of these criteria: visual, hearing or orthopedic impairment; other neurological disorders, dementia, severe dyskinesia, medical diagnosis of dementia, past-history or presence of psychotic episode reported in the medical file since the PD diagnosis, implantable deep brain stimulator.”

R1 - Comment#4

Could the authors provide further rationale for why they tested off medication and describe how this may have impacted on findings?

Authors:

Page 3, Lines 57-61: “To investigate the effects of a cognitive dual task on balance performance in pwPD, the medication state must also be considered. Levodopa’s effect on static balance remains controversial [12,13]. Both increase and decrease of the sway area during quiet standing have been reported after levodopa intake during quiet standing has been reported to both [14]. While understanding the differences between the ON and OFF-medication states is very important for follow-up in PD, the investigation of differences between pwPD with low balance impairment during their OFF-medication state and controls could highlight balance-specific PD markers, which potentially help to identify pwPD who are still not diagnosed.”

R1 - Comment#5

The findings around dual task are hard to interpret. The findings may be due to the mild disease of the participants, or perhaps cortical control isn’t as important in early disease or compared to other motor tasks such as gait. One recent review suggests the importance of cerebellum for postural control (https://doi.org/10.1016/j.neubiorev.2020.04.028). It has been noted in the limitations that cognitive performance during the dual task was not assessed and therefore the trade-off between cognitive and balance tasks cannot be determined.

Authors: Thank you for this comment, both for clear verbiage and reference. We added this into the limitation section. Please see Page 13, Lines 249-256: “Some limitations must be discussed. The findings around dual-task are hard to interpret. First, the cognitive performance during the dual-task was not assessed and the trade-off between cognitive and balance tasks cannot be determined. Then, the findings may be due to the mild disease of the participants, or perhaps cortical control is not as important in early disease or compared to other motor tasks such as gait. One recent review suggests the importance of cerebellum for postural control [52]. Regarding the group of controls, we did not compare pwPD to age-matched adults. Due to current restriction (i.e. COVID-19), the access to healthy participants was highly limited. This was overcome by measuring young adults instead, in a different laboratory, and it also explained the different force plate and EMG equipment. However, the data analysis was conducted by the same person using the same algorithm. For this reason, the results from the control group were used here as a reference for an optimal functioning, to see if the trends were the same under blinded condition and/or dual-task, but not for a direct comparison with the pwPD. Finally, as mentioned previously, the use of a single, large force-plate did not allow to investigate further differences between the forward and backward legs.

Future studies should first overcome these limitations. The use of a conceptual model would be helpful to characterize patterns of cognitive-motor dual-task interference [53], especially when, as suggested by Workman & Thrasher [43], some symptoms in PD might simulate stability in some posturographic measures and affect interpretation.”

.

R1 - Comment#6

Statistically there are a lot of multiple comparisons alongside a small number of participants, the authors should be more stringent and control for multiple comparisons.

Authors: Multiple regressions have been initially run with age, sex, UPDRS and Berg Balance Scale as potential confounders. As no effects were observed, we only reported the absence of effects. Please see the table 1 completed. Age was obviously different; height was lower in pwPD (p=0.002) and weight was not different (p=0.71). BMI was not statistically different (p=0.08).

Reviewer 2 Report

Summary

The authors investigated the effects of visual deprivation and dual-tasking on tandem standing balance in PwPD and young healthy controls, using both linear and non-linear measures. They found significant effects in eyes closed vs. eyes open conditions for several COP measures, a few EMG measures, and little change in the non-linear analyses. Interestingly, there were few, if any, dual-task effects. Some work on the presentation of the Results and bolstering the Discussion would improve the overall quality of this manuscript.

Overall comment

Although it is clear this was written by a non-native English speaker, the flow is rarely disrupted by poor English and does not distract from the message. Nevertheless, I will note one obvious spelling error. I also recommend running an additional spell check ensuring the software is set to the appropriate English setting for the journal (American or British).

Title

  • Change “stabilising" to “stabilizing”

Abstract

  • Results: Clarify that the first portion/sentence of the results is for PwPD

Introduction

Major comments:

  • Please justify your choice to test only if the OFF state, either in the Introduction or in the Methods.
  • Line 60: I see a purpose statement/summary. Did you have any a priori hypotheses? These should be mentioned here.

Minor comments:

  • Line 55: There is not a closing parenthesis “)” to go with the opening parenthesis “(“

Materials and Methods

Major comments: Section 2.1 Subjects should be reorganized to improve the flow. Specific suggestions below.

  • Line 63: Please justify your choice for using young healthy subjects as controls and not age-matched, or at least similarly aged, adults.
  • Lines 65-66: The UPDRS and H&Y being performed by an experienced neuro-physiotherapist should go at the end of the paragraph.
  • Lines 70-74: These inclusion criteria should be listed with the others (Line 63-65). Thus the flow will be the 1) existing inclusion criteria, 2) then these on 70-74, 3) and then mentioning the neuro-physiotherapist.
  • Lines 79-82: How did you determine foot dominance? Did you consider assigning stances according to more- and less-PD affected sides instead of dominant and non-dominant for the PwPD?
  • Lines 92-95: Please justify your use of serial 3 subtraction for the secondary/cognitive task. You mentioned serial 7 subtraction in the Introduction, so there is a disconnect between these two sections. Similarly, did you consider the effect of articulation (speaking) on your COP measures (see Dault, et al. (2003))? If you can’t provide evidence that speaking did NOT affect your COP outcomes, I recommend adding this to the Discussion or to the limitations.

M.C. Dault, L. Yardley, J.S. Frank, Does articulation contribute to modifications of postural control during dual-task paradigms? Cogn. Brain Res. 16 (2003) 434–440.)

Minor comments:

  • Line 98-99: Why did the PwPD and the control subjects use different force plates and EMG equipment? I list this as “minor” because it’s not problematic for your outcomes, but if you can state a reason it would make your methodology clearer.

Results

Major comments:

  • Lines 148-149: Please add the variability (SD) to the mean data in the text.
  • Table 2: Can you indicate significant differences within the table? E.g., * = significantly different from EO-ST, # = significantly different from EC-ST, etc. Unless otherwise specified by the journal, the abbreviations should be put under the table in the note/footer section, which is also where you define your symbols used to indicate significant differences.
  • Table 3: Same comment on abbreviations as above. In addition, this table is a long and rather cumbersome. It could be improved by separating each major section into individual tables (PD COP measures, control COP measures, PD EMG measures, control EMG measure, PD non-linear, control non-linear). Also, many of the rows don’t add to your overall story (e.g., COP frequency measures, most of the entropy results). You could remove these from the tables and mention significant differences between the groups in the text. Again, this would simplify your table(s) and improve data/results readability.

Minor comments:

  • Lines 134-135: Much of this information is also found in the table. To simplify, I recommend removing most of this text and just referring the readers to the demographics table.

Discussion

Major comments:

  • Line 262: You mention a greater reliance on “the integration of available sensory outputs” but make no mention of what these might be. For balance, the three systems primarily involved are the vision (which you discuss), vestibular, and somatosensory systems. Please briefly discuss the role/contribution of vestibular and somatosensory to balance to add more context to your discussion of vision.
  • Lines 278-281: You mention the Nieuwhof study, but what about other studies with designs similar to yours? For example, Workman and Thrasher (2019) found DT and eyes-open/eyes-closed effects both ON and OFF balance performance, which would be interesting to mention in your discussion as well. Especially when you talk about the effects of dopaminergic medications, either in this paragraph or in your limitations paragraph.

Workman CD, and Thrasher TA (2019). The Influence of Dopaminergic Medication on Balance Automaticity in Parkinson's Disease. Gait Posture 70, 98-103.

  • Lines 288-291: You mention one of the two most popular DT theories, (Bottleneck) but do not discuss the other popular theory (Capacity Sharing). (See Plummer and Eskes (2015) for an overview of both theories.) Your Discussion would benefit from including the other theory and mentioning if/how your data support one or the other.
  1. Plummer, G. Eskes, Measuring treatment effects on dual-task performance: a

framework for research and clinical practice, Front. Hum. Neurosci. 9 (2015) 225.

Minor comments:

  • You may need to add to the limitations, as per my comments above. I also recommend adding some discussion about future studies. What can others take from your study to improve their next study design? What did you learn from yours that you can share with others to improve this niche of science? Can rehabilitation therapists use this information? You don’t need to answer all of my questions, but a future studies paragraph will help finalize your Discussion.

Author Contributions

You should delete the descriptive text on how to fill out this section “For research articles… should be used”

Conflicts of Interest

You have two contradictory statements here. The first sentence is that there are no conflicts to declare, but then the second sentence is that Dr. Gouelle is employed by ProtoKinetics. Please clarify.

Author Response

Reviewer 2

Summary

The authors investigated the effects of visual deprivation and dual-tasking on tandem standing balance in PwPD and young healthy controls, using both linear and non-linear measures. They found significant effects in eyes closed vs. eyes open conditions for several COP measures, a few EMG measures, and little change in the non-linear analyses. Interestingly, there were few, if any, dual-task effects. Some work on the presentation of the Results and bolstering the Discussion would improve the overall quality of this manuscript.

Overall comment

Although it is clear this was written by a non-native English speaker, the flow is rarely disrupted by poor English and does not distract from the message. Nevertheless, I will note one obvious spelling error. I also recommend running an additional spell check ensuring the software is set to the appropriate English setting for the journal (American or British).

R2 - Comment#1

Change “stabilising" to “stabilizing”.

R2 - Comment#2

Abstract / Results: Clarify that the first portion/sentence of the results is for PwPD.

Authors: It has been done. Please see the modification on the abstract line 24-35:

Methods: Fifteen OFF-medication state pwPD (9 women and 6 men), 67.7±7.3 years old, diagnosed PD since 5.4±3.4 years, only Hoehn and Yahr state 2 and fifteen young control adults (7 women and 8 men) aged 24.9±4.9 years, performed semi-tandem task under four randomized experimental conditions: eyes opened single-task, eyes closed single-task, eyes opened dual-task and eyes closed dual-task. The center of pressure (COP) was measured using a force plate and electromyography signals (EMG) of the ankle/hip muscles were recorded. Traditional parameters, including COP pathway length, ellipse area, mediolateral/anteroposterior root-mean-square and non-linear measurements were computed. The effect of vision privation, cognitive task, and vision X cognitive was investigated by a 2 (eyes opened/eyes closed) x 2 (postural task alone/with cognitive task) repeated-measures ANOVA after application of a Bonferroni pairwise correction for multiple comparisons. Significant interactions were further analyzed using post-hoc tests. Results: In pwPD, both COP pathway length (p<0.01), ellipse area (p<0.01) and mediolateral/anteroposterior root-mean-square (p<0.01) were increased with the eyes closed, while the dual-task had no significant effect when compared to the single-task condition. Comparable results were observed in the control group for who COP pathway was longer in all conditions compared to eyes opened single-task (p<0.01) and longer in conditions with eyes closed compared to eyes opened dual-task (p<0.01). Similarly, all differences in EMG activity of pwPD were exclusively observed between eyes opened versus eyes closed conditions, and especially for the forward leg’s soleus (p<0.01) and backward tibialis anterior (p<0.01). “

R2 - Comment#3

Introduction / Major comments: Please justify your choice to test only if the OFF state, either in the Introduction or in the Methods.

Authors: This is an important question as also highlight Reviewer1’s comment 4.

Page 3, Lines 57-61: “To investigate the effects of a cognitive dual task on balance performance in pwPD, the medication state must also be considered. Levodopa’s effect on static balance remains controversial [12,13]. Both increase and decrease of the sway area during quiet standing have been reported after levodopa intake during quiet standing has been reported to both [14]. While understanding the differences between the ON and OFF-medication states is very important for follow-up in PD, the investigation of differences between pwPD with low balance impairment during their OFF-medication state and controls could highlight balance-specific PD markers, which potentially help to identify pwPD who are still not diagnosed.”

R2 - Comment#4

Line 60: I see a purpose statement/summary. Did you have any a priori hypotheses? These should be mentioned here.

Authors: We have added a priori hypotheses.

Please refer to Page 3, Lines 67-69: “We expected an effect of both visual deprivation and dual-task on the semi-tandem performances, especially in the patients. We hypothesized a significant increase in neuromuscular activity of the leg muscles during the more challenging condition, in relation with an increase of ankle contribution to maintain body sways [18].”

R2 - Comment#5

Minor comments:

Line 55: There is not a closing parenthesis “)” to go with the opening parenthesis “(“

Authors: Thank you for finding this typo, we correct it.

R2 - Comment#6

Materials and Methods

Major comments: Section 2.1 Subjects should be reorganized to improve the flow. Specific suggestions below.

Line 63: Please justify your choice for using young healthy subjects as controls and not age-matched, or at least similarly aged, adults.

Authors: The authors are aware of limitations in the choice of controls, which is due to current pandemic situation. We addressed it in the limitation.

Page 14, Lines 252-256: “Regarding the group of controls, we did not compare pwPD to age-matched adults. Due to current restriction (i.e. COVID-19), the access to healthy participants was highly limited. This was overcome by measuring young adults instead, in a different laboratory, and it also explained the different force plate and EMG equipment. However, the data analysis was conducted by the same person using the same algorithm. For this reason, the results from the control group were used here as a reference for an optimal functioning, to see if the trends were the same under blinded condition and/or dual-task, but not for a direct comparison with the pwPD.”

R2 - Comment#7

Lines 65-66: The UPDRS and H&Y being performed by an experienced neuro-physiotherapist should go at the end of the paragraph.

Authors: We updated the section 2.1. Subject and 2.2 Experiment.

R2 - Comment#8

Lines 70-74: These inclusion criteria should be listed with the others (Line 63-65). Thus the flow will be the 1) existing inclusion criteria, 2) then these on 70-74, 3) and then mentioning the neuro-physiotherapist.

Authors: We updated the section 2.2. Experiment. Please see these updated sections on Pages 4 and 5.

R2 - Comment#9

Lines 79-82: How did you determine foot dominance? Did you consider assigning stances according to more- and less-PD affected sides instead of dominant and non-dominant for the PwPD?

Authors:

Page 4, Lines 90-91: “The foot dominance [25] has been determined with the mobilizing or manipulating limb as the dominant foot, and the supporting limb as the non-dominant limb.”

R2 - Comment#10

Lines 92-95: Please justify your use of serial 3 subtraction for the secondary/cognitive task. You mentioned serial 7 subtraction in the Introduction, so there is a disconnect between these two sections. Similarly, did you consider the effect of articulation (speaking) on your COP measures (see Dault, et al. (2003))? If you can’t provide evidence that speaking did NOT affect your COP outcomes, I recommend adding this to the Discussion or to the limitations.

M.C. Dault, L. Yardley, J.S. Frank, Does articulation contribute to modifications of postural control during dual-task paradigms? Cogn. Brain Res. 16 (2003) 434-440.).

Authors: Remarkably interesting comment. We changed the introductive example with 3, even if it was only one example. We did not want to propose a dual task to complex and decided to use 3, however as discussed in this manuscript, the absence of real DT effect might be due to a task which was too easy for our participants.

Because the UPDRS III has been assessed, we obviously scored the first item which is related to the speech. No participant had speech problems (all participants with 0 for this item). We added this information within the table 1 and in the section 2.2. Experiment.

Page 4, Lines 146-147: “A Score of 0 for the first item of UPDRS III, which is related to speech, provided evidence that the participants had no issue with articulation that would contribute to modification in COP measures [29].”

R2 - Comment#11

Minor comments:

Line 98-99: Why did the PwPD and the control subjects use different force plates and EMG equipment? I list this as “minor” because it’s not problematic for your outcomes, but if you can state a reason it would make your methodology clearer.

Authors:

Page 14, Lines 252-256: “Regarding the group of controls, we did not compare pwPD to age-matched adults. Due to current restriction (i.e. COVID-19), the access to healthy participants was highly limited. This was overcome by measuring young adults instead, in a different laboratory, and it also explained the different force plate and EMG equipment. For this reason, the results from the control group were used here as a reference for an optimal functioning, to see if the trends were the same under blinded condition and/or dual-task, but not for a direct comparison with the pwPD.”

R2 - Comment#12

Results / Major comments:

Lines 148-149: Please add the variability (SD) to the mean data in the text.

Authors: We have added SD. Please see line 149-150: “(p<0.001, longer with eyes closed: EO-ST 1606±1060 mm versus EC-ST 2546±930 mm, EO-DT 2105±1749 mm versus EC-DT 2874±1776 mm), ellipse area (p<0.001, larger with EC: EO-ST 101±56 mm² versus EC-ST 283±168 mm², EO-DT 159±118 mm² versus EC-DT 296±177 mm²)“

And please see line 156: “(p<0.01, EO-ST 1654±220 mm, EC-ST 1985±223 mm, EO-DT 1793±272 mm, EC-DT 2041±332 mm) “

R2 - Comment#13

Table 2: Can you indicate significant differences within the table? E.g., * = significantly different from EO-ST, # = significantly different from EC-ST, etc. Unless otherwise specified by the journal, the abbreviations should be put under the table in the note/footer section, which is also where you define your symbols used to indicate significant differences.

Authors: Table 2 has been updated according this advise and now included symbols for significant differences. Abbreviation have been moved under the tables.

R2 - Comment#14

Table 3: Same comment on abbreviations as above. In addition, this table is a long and rather cumbersome. It could be improved by separating each major section into individual tables (PD COP measures, control COP measures, PD EMG measures, control EMG measure, PD non-linear, control non-linear). Also, many of the rows don’t add to your overall story (e.g., COP frequency measures, most of the entropy results). You could remove these from the tables and mention significant differences between the groups in the text. Again, this would simplify your table(s) and improve data/results readability.

Authors: Non-significant results have been removed from Table 3, which simplify presentation of the results.

R2 - Comment#15

Minor comments:

Lines 134-135: Much of this information is also found in the table. To simplify, I recommend removing most of this text and just referring the readers to the demographics table.

Authors: We refer now to the table 1.

Page 7, Line 145: “The characteristics of the pwPD and participants in the control group are displayed in Table 1.”

R2 - Comment#16

Discussion / Major comments: Line 262: You mention a greater reliance on “the integration of available sensory outputs” but make no mention of what these might be. For balance, the three systems primarily involved are the vision (which you discuss), vestibular, and somatosensory systems. Please briefly discuss the role/contribution of vestibular and somatosensory to balance to add more context to your discussion of vision.

Authors: We agree with this comment. We do not discuss extensively but we added specification about the components we were referring to and a new reference.

Page 12, Lines 191-193: “A healthy person relies on visual, somatosensory, and vestibular information for balance control, but all or some of the components of this system may be dysfunctional in Parkinsonian patients [33].”

R2 - Comment#17

Lines 278-281: You mention the Nieuwhof study, but what about other studies with designs similar to yours? For example, Workman and Thrasher (2019) found DT and eyes-open/eyes-closed effects both ON and OFF balance performance, which would be interesting to mention in your discussion as well. Especially when you talk about the effects of dopaminergic medications, either in this paragraph or in your limitations paragraph.

Workman CD, and Thrasher TA (2019). The Influence of Dopaminergic Medication on Balance Automaticity in Parkinson's Disease. Gait Posture 70, 98-103.

Authors: Thank you for this reference we were unfortunately not aware before. The authors did a great work in their discussion and we would not like to paraphrase their explanation. We simply added few lines to cite their work and explain what their findings were when looking at dual task difference between ON/OFF states.

Page 13, Lines 217-220: “Based on recent results by Workman & Thrasher [43], it would not be the case. During a similar protocol to ours, sixteen pwPD completed single- and dual-task standing with eyes open and eyes closed for 3 min each in off and on medication states, to investigate if dopaminergic medication improved standing balance automaticity during a phoneme monitoring dual-task. No change suggesting an improvement in automaticity was found during dual-task and a negative effect was even seen under medication.”

R2 - Comment#18

Lines 288-291: You mention one of the two most popular DT theories, (Bottleneck) but do not discuss the other popular theory (Capacity Sharing). (See Plummer and Eskes (2015) for an overview of both theories.) Your Discussion would benefit from including the other theory and mentioning if/how your data support one or the other.

Plummer, G. Eskes, Measuring treatment effects on dual-task performance: a framework for research and clinical practice, Front. Hum. Neurosci. 9 (2015) 225.

 Authors: Please see Page 13, Lines 221-228.

“Dual task paradigms are undertaken to challenge attentional capacities. Two main theorical models propose to rationalize the mechanisms which induce interferences in one or both tasks. The capacity sharing model considers attention as a limited resource and implies a sharing of attention between the two tasks. If both tasks requirement overpasses the attentional capacity, then the achievement of the primary task must be done to the detriment of the secondary task by drawing on the attentional load that it brings into play [44]. On the other hand, the bottleneck theory [45] suggests that interference occurs when two tasks compete for the access to an attentional mechanism whose capacity is limited to the management of a single central operation at a time. To complete one task, processing of the second task is temporarily postponed, resulting in performance decrements in the second task. Nonappearance of difference between single task and dual task could be explained by an insufficiently challenging cognitive task for attentional resources of our participants (capacity sharing) or because the second task was not in direct competition with the standing task (bottleneck).”

R2 - Comment#19

Minor comments: You may need to add to the limitations, as per my comments above. I also recommend adding some discussion about future studies. What can others take from your study to improve their next study design? What did you learn from yours that you can share with others to improve this niche of science? Can rehabilitation therapists use this information? You don’t need to answer all of my questions, but a future studies paragraph will help finalize your Discussion.

Authors: In addition to major revision of the limitation, a sentence introduces now future perspectives.

Page 14, Lines 258-260: “Future studies should first overcome these limitations. The use of a conceptual model would be helpful to characterize patterns of cognitive-motor dual-task interference [53], especially when, as suggested by Workman & Thrasher [43], some symptoms in PD might simulate stability in some posturographic measures and affect interpretation.”

R2 - Comment#20

Author Contributions

You should delete the descriptive text on how to fill out this section “For research articles… should be used”.

Authors: Thank you for highlighting this omission. We delete it.

R2 - Comment#21

Conflicts of Interest

You have two contradictory statements here. The first sentence is that there are no conflicts to declare, but then the second sentence is that Dr. Gouelle is employed by ProtoKinetics. Please clarify.

Authors: We have removed this sentence. Dr Gouelle is employed by ProtoKinetics, but this does not represent a conflict of interest here, due either to the methodology or to the purpose of the study.

Reviewer 3 Report

The manuscript addresses an interesting and relevant issue in the rehabilitation of people with Parkinson Disease (pwPD). However, methodological issues statistical analysis details and results presentation must be improved to justify discussion and conclusion. The major concerns are related to the type of cognitive dual-task, statistics and the healthy control group. The dual-task condition most probably was not challenging for the patients and it would strongly influence results. The Authors did not consider "group" as a between-group factor and they did not report the overall statistics before post-hoc comparisons. It limits the data interpretation. Tables are very difficult to read. They must be condensed and clearly reported. The healthy control group was younger than the pwPD group. No comparisons between groups have been reported in demographic and clinical data.

Abstract. It must be improved including details on demographic and clinical data such as H&Y stage (to justify the absence of postural control deficits), mean age of pwPD and healthy controls; methods and statistics reporting should be improved. Results must be reported more in details including p values (when applicable).

Introduction. The aim of the study is not clearly reported as well as the authors' hypothesis are missing.

Methods. Inclusion criteria should be reported highlighting the H&Y stage which is crucial to include patients without postural instability. Please, report the side more affected and if it influenced the leg position on the platform. 

Statistics. "group" as a between-group factor must be included in the model. Comparisons of demographic and clinical data are not reported.

Results. The overall statistical significance for each outcome is not reported as well as the interactions significances have not been reported. Thus, it is difficult to interpret post-hoc comparisons. Tables are very difficult to read and understand. The two groups were significantly different on age and it would limit the data interpretation.

Discussion and conclusion might be justified if results were reported more in details. The type of dual-task used, as the authors stated, might limit results and then data interpretation.

Author Response

Reviewer 3

Comments and Suggestions for Authors

The manuscript addresses an interesting and relevant issue in the rehabilitation of people with Parkinson Disease (pwPD). However, methodological issues statistical analysis details and results presentation must be improved to justify discussion and conclusion. The major concerns are related to the type of cognitive dual-task, statistics and the healthy control group. The dual-task condition most probably was not challenging for the patients and it would strongly influence results. The Authors did not consider "group" as a between-group factor and they did not report the overall statistics before post-hoc comparisons. It limits the data interpretation. Tables are very difficult to read. They must be condensed and clearly reported. The healthy control group was younger than the pwPD group. No comparisons between groups have been reported in demographic and clinical data.

R3 - Comment#1

Abstract. It must be improved including details on demographic and clinical data such as H&Y stage (to justify the absence of postural control deficits), mean age of pwPD and healthy controls; methods and statistics reporting should be improved. Results must be reported more in details including p values (when applicable).

Authors:

Abstract has been updated.

Please see the modification on the abstract line 24-35:

Methods: Fifteen OFF-medication state pwPD (9 women and 6 men), 67.7±7.3 years old, diagnosed PD since 5.4±3.4 years, only Hoehn and Yahr state 2 and fifteen young control adults (7 women and 8 men) aged 24.9±4.9 years, performed semi-tandem task under four randomized experimental conditions: eyes opened single-task, eyes closed single-task, eyes opened dual-task and eyes closed dual-task. The center of pressure (COP) was measured using a force plate and electromyography signals (EMG) of the ankle/hip muscles were recorded. Traditional parameters, including COP pathway length, ellipse area, mediolateral/anteroposterior root-mean-square and non-linear measurements were computed. The effect of vision privation, cognitive task, and vision X cognitive was investigated by a 2 (eyes opened/eyes closed) x 2 (postural task alone/with cognitive task) repeated-measures ANOVA after application of a Bonferroni pairwise correction for multiple comparisons. Significant interactions were further analyzed using post-hoc tests. Results: In pwPD, both COP pathway length (p<0.01), ellipse area (p<0.01) and mediolateral/anteroposterior root-mean-square (p<0.01) were increased with the eyes closed, while the dual-task had no significant effect when compared to the single-task condition. Comparable results were observed in the control group for who COP pathway was longer in all conditions compared to eyes opened single-task (p<0.01) and longer in conditions with eyes closed compared to eyes opened dual-task (p<0.01). Similarly, all differences in EMG activity of pwPD were exclusively observed between eyes opened versus eyes closed conditions, and especially for the forward leg’s soleus (p<0.01) and backward tibialis anterior (p<0.01). “

R3 - Comment#2

Introduction. The aim of the study is not clearly reported as well as the authors' hypothesis are missing.

Authors: We have added a priori hypotheses. Please refer to Page 3, Lines 66-69: “This study was set to test how visual deprivation and dual-task affect body sway in OFF-state pwPD who hold a semi-tandem stance. A control group of healthy subjects underwent the same protocol. We expected an effect of both visual deprivation and dual-task on the semi-tandem performances, especially in the patients. We hypothesized a significant increase in neuromuscular activity of the leg muscles during the more challenging condition, in relation with an increase of ankle contribution to maintain body sways [18].”

R3 - Comment#3

Methods. Inclusion criteria should be reported highlighting the H&Y stage which is crucial to include patients without postural instability. Please, report the side more affected and if it influenced the leg position on the platform.

Authors:

Inclusion/Exclusion section has been written in an improved manner.

Page 3, Line 73-79: “The inclusion criteria for the pwPD were as follows: original diagnosis of PD by a neurologist based on standard clinical criteria [19]; to be able to walk without walking aids (i.e. without a walker or a cane); Hoehn and Yahr [20] state 2 only (bilateral involvement without impairment of balance); no fall history; no antiparkinsonian medications taken for 12 hours prior to the study session (i.e. “OFF-medication state”); Mini-Mental State Examination [21] score above 26, which considered as the cut-off for normal cognitive function. Restriction to pwPD with Hoehn and Yahr at state 2 was set to involve persons who did not demonstrate impairment of the static balance. Patients were not included if they presented any of these criteria: visual, hearing or orthopedic impairment; other neurological disorders, dementia, severe dyskinesia, medical diagnosis of dementia, past-history or presence of psychotic episode reported in the medical file since the PD diagnosis, implantable deep brain stimulator.”

Foot dominance has been addressed.

Page 4, Lines 90-91: “The foot dominance [25] has been determined with the mobilizing or manipulating limb as the dominant foot, and the supporting limb as the non-dominant limb.”

R3 - Comment#4

Statistics. "group" as a between-group factor must be included in the model. Comparisons of demographic and clinical data are not reported.

Authors: We have added the results for comparison of demographics in table 1. Age was obviously different; height was lower in pwPD (p=0.002) and weight was not different (p=0.71). BMI was not statistically different (p=0.08).

In the limitation section, we explain the reason for unmatched controls which lead us to not run direct comparison between groups.

Page 14, Lines 252-256: “Regarding the group of controls, we did not compare pwPD to age-matched adults. Due to current restriction (i.e. COVID-19), the access to healthy participants was highly limited. This was overcome by measuring young adults instead, in a different laboratory, and it also explained the different force plate and EMG equipment. For this reason, the results from the control group were used here as a reference for an optimal functioning, to see if the trends were the same under blinded condition and/or dual-task, but not for a direct comparison with the pwPD.”

R3 - Comment#5

Results. The overall statistical significance for each outcome is not reported as well as the interactions significances have not been reported. Thus, it is difficult to interpret post-hoc comparisons. Tables are very difficult to read and understand. The two groups were significantly different on age and it would limit the data interpretation.

Authors: Major revision has been done in all sections of this manuscript on the base of the reviewers’ comments and advises. We expect that the edits we made should satisfy your requests.

R3 - Comment#6

Discussion and conclusion might be justified if results were reported more in details. The type of dual-task used, as the authors stated, might limit results and then data interpretation.

Authors: As per similar comments by Reviewer 2, we improved the discussion about dual task.

Page 13, Lines 221-228:

“Dual task paradigms are undertaken to challenge attentional capacities. Two main theorical models propose to rationalize the mechanisms which induce interferences in one or both tasks. The capacity sharing model considers attention as a limited resource and implies a sharing of attention between the two tasks. If both tasks requirement overpasses the attentional capacity, then the achievement of the primary task must be done to the detriment of the secondary task by drawing on the attentional load that it brings into play [44]. On the other hand, the bottleneck theory [45] suggests that interference occurs when two tasks compete for the access to an attentional mechanism whose capacity is limited to the management of a single central operation at a time. To complete one task, processing of the second task is temporarily postponed, resulting in performance decrements in the second task. Nonappearance of difference between single task and dual task could be explained by an insufficiently challenging cognitive task for attentional resources of our participants (capacity sharing) or because the second task was not in direct competition with the standing task (bottleneck).”

As well at Page 14, Lines 249-250:

“The findings around dual task are hard to interpret. First, the cognitive performance during the dual task was not assessed and the trade-off between cognitive and balance tasks cannot be determined.”

Round 2

Reviewer 2 Report

The authors have addressed my major concerns and I find this version of the manuscript publishable.

Author Response

Thank you from all of the authors 

Reviewer 3 Report

The Authors did not fully address concerns related to statistics and results presentation. The Authors did not consider in the model "group" as a between-group factor, and they did not report the overall statistics results before post-hoc comparisons. It limits the data interpretation: mixed ANOVA with the task (ST, DT) and condition (eyes open or eyes closed) as within-subject factors and group (HS or PD) as between-subject factor.  Results must be reported also in terms of interactions such as  task*group, condition*grop and task*condition*group reporting F(dftime, dferror) = F-value, p = p-value.

Tables are very difficult to read. Table 1 does not report the p-value for all the variables. Table 2 and 3 must be improved and condensed in a single table. Table 3 is just the statistics output with redundant information.

Author Response

Reviewer 3 (Review 2)

 Comment #1

The Authors did not fully address concerns related to statistics and results presentation. The Authors did not consider in the model "group" as a between-group factor, and they did not report the overall statistics results before post-hoc comparisons. It limits the data interpretation: mixed ANOVA with the task (ST, DT) and condition (eyes open or eyes closed) as within-subject factors and group (HS or PD) as between-subject factor. Results must be reported also in terms of interactions such as  task*group, condition*grop and task*condition*group reporting F(dftime, dferror) = F-value, p = p-value.

Authors:

We highly appreciate your comment and agree that it should be made if we were comparing the groups. As we underlined in the limitation section, the controls were unmatched and not used for direct comparison with the PD group. As we tried to answer the reviewers’ comments and especially to your comment #4, we had added to the Table 1 between-group comparison to highlight differences in the demographic features. Maybe this may lead readers to think that we compare groups.

However, there is no information later in the manuscript nor tables about any direct comparison. With the limitation section explaining why, it seems there are enough elements for the reader to understand the situation.

Therefore, regarding “group” as a between-group factor, there is no reason to run such a statistics.

Please see line 252 to 256: “Regarding the group of controls, we did not compare pwPD to age-matched adults. Due to current restriction (i.e. COVID-19), the access to healthy participants was highly limited. This was overcome by measuring young adults instead, in a different laboratory, and it also explained the different force plate and EMG equipment. However, the data analysis was conducted by the same person using the same algorithm. For this reason, the results from the control group were used here as a reference for optimal functioning, to see if the trends were the same under blinded condition and/or dual-task, but not for a direct comparison with the pwPD.

Comment #2

Tables are very difficult to read. Table 1 does not report the p-value for all the variables. Table 2 and 3 must be improved and condensed in a single table. Table 3 is just the statistics output with redundant information.

Authors:

For Table 1, we added p-value for age, gender, and dominant leg. However, as answered in comment #1, in case it is misleading for readers, we would accept to remove this column about between group comparison.

For Table 3, we acknowledge that it is pretty like Table 2 since we added symbols according to Reviewer 2’s request. A reviewer thought it was important to provide this detailed information. Anyway, our goal was to give results from the control group to see the trends.

To answer the current comment, we removed results about participants with PD in Table 3 and present only significant results for the control group.
